# Impact of Aging Microbiome on Metabolic Profile of Natural Aging Huangjiu through Machine Learning

**DOI:** 10.3390/foods12040906

**Published:** 2023-02-20

**Authors:** Huakun Yu, Shuangping Liu, Zhilei Zhou, Hongyuan Zhao, Yuezheng Xu, Jian Mao

**Affiliations:** 1National Engineering Research Center for Cereal Fermentation and Food Biomanufacturing, State Key Laboratory of Food Science and Technology, School of Food Science and Technology, Jiangnan University, Wuxi 214122, China; 2Jiangsu Provincial Research Center for Bioactive Product Processing Technology, Jiangnan University, Wuxi 214122, China; 3Jiangnan University (Shaoxing) Industrial Technology Research Institute, Shaoxing 312000, China; 4School of Artificial Intelligence and Computer Science, Jiangnan University, Wuxi 214122, China; 5National Engineering Research Center of Huangjiu, Zhejiang Guyuelongshan Shaoxing Wine Co., Ltd., Shaoxing 312000, China

**Keywords:** aging microbiome, metabolic profiling, multi-omics, machine learning, aging regulation, fermented alcoholic beverages

## Abstract

Aging is a time-consuming step in the manufacturing of fermented alcoholic beverages. Natural-aging huangjiu sealed in pottery jars was taken as an example to investigate the changes of physiochemical indexes during aging and to quantify intercorrelations between aging-related factors and metabolites through machine learning methods. Machine learning models provided significant predictions for 86% of metabolites. Physiochemical indexes well reflected the metabolic profile, and total acid was the most important index that needed to be controlled. For aging-related factors, several aging biomarkers of huangjiu were also well predicted. Feature attribution analysis showed aging year was the most powerful predictive factor, and several microbial species were significantly associated with aging biomarkers. Some of the correlations, mostly connected to environmental microorganisms, were newly found, showing considerable microbial influence on aging. Overall, our results reveal the potential determinants that affect the metabolic profile of aged huangjiu, paving the way for a systematical understanding of changes in metabolites of fermented alcoholic beverages.

## 1. Introduction

As one of the time-honored drinks, fermented alcoholic beverages are renowned for the distinct flavor, which is derived from aging to a great extent. Newly-produced wines have unharmonious aromas and unpleasant tastes, requiring long-term storage for substance transformation to form the characteristic aroma profile. In a typical fermented alcoholic beverage sample, water and alcohol make up over 95% of the total contents, and the remaining 5% are the main flavor contributors [1,2]. The process by which flavor compounds such as acids, alcohols, aldehydes, and esters mutually transform under specific conditions and eventually reach the balance of various microconstituents is known as aging of fermented alcoholic beverages [3,4]. The resulting sensory perception of aged products depends on a number of factors, including duration, environmental conditions (temperature and humidity), initial physiochemical parameters (pH, total acid, and alcohol level), and the type of aging containers.

Compared with distilled liquor, fermented alcoholic beverages are relatively low in alcohol content (<20% Alcohol By Volume (ABV)) and rich in nutrients such as amino acids and saccharides, and aging containers are suitable habitats for some microorganisms, including fungi and bacteria [1,5,6], which possibly influence the sensory properties of wine body. These microorganisms may originate from the fermentation process or may have remained in the wine containers after manufacture or previous usage. During the process of aging and maturation, microorganisms such as lactic acid bacteria (LAB) and acetic acid bacteria (AAB) produce a wide range of metabolites such as organic acids [7,8]. While a moderate amount of acids improves the flavor properties, excessive organic acids are harmful and unwanted, which is often associated with spoilage. LAB and AAB, which originate from the fermentation process, are the most-studied microorganisms during aging for acid-producing ability [8,9,10,11]. Roos et al. [12] used a highly selective medium and matrix-assisted laser desorption ionization–time of flight mass spectrometry (MALDI-TOF MS) as a high-throughput dereplication method combined with comparative gene sequencing for the isolation and identification of AAB in lambic beer production. AAB such as *Acetobacter orientalis* and *Acetobacter pasteurianus* exist during the maturation process. LAB, identified as the main cause of rancidity, was also isolated from aged huangjiu samples [8,13]. While the harmfulness of LAB and AAB during aging was investigated, the function of other aging microorganisms, especially those from the environment, was usually ignored. Previous aging research has concentrated on the physiochemical reactions [4,14,15]. A comprehensive understanding of aging’s microbial influence enables brewers to prevent spoilage and improve the aging process.

To analyze the microbial mechanisms of the complex aging process of fermented alcoholic beverages systematically and comprehensively, it takes more than single metabolomics. Combining omics data from multiple analytical methods is a promising approach for gaining a comprehensive understanding of the correlation and interaction between various omics [16,17,18], and machine learning methods are the most recent data mining methods for multi-omics datasets [19,20,21]. Machine learning pipelines have been shown to be an efficient and precise method for dealing with gut microbiome. As machine learning models are generally regarded as hard-to-understand “black boxes”, explainable artificial intelligence (XAI) [22,23,24], a set of math techniques that help humans fully understand the decision-making process of the model, has thus received great attention in safety-sensitive tasks such as food manufacturing and disease diagnosis. Chen et al. [25] developed an ensemble learning model with multiple heterogeneous algorithms and combined species and pathway profiles for a multi-view learning of people’s age. Furthermore, the model was interpreted using permutation feature importance, and several potential age-related biomarkers were identified. Bar et al. [26] identified the metabolites in the human serum samples of healthy volunteers and used LightGBM to predict more than 1000 unique metabolites in the samples based on the data of the gut microbiome and host genetics. Then, feature attribution analysis was applied to reveal the potential determinants of major metabolites, thus better helping us to understand the metabolism mechanism and to develop interventions. Though there were few studies that correlated metabolites in huangjiu with related microbiomes using machine learning and XAI, these methods have shown powerful data management functions and data mining abilities in the analysis of multi-omics.

Among fermented alcoholic beverages, huangjiu has a relatively high alcohol (16–18% ABV) and acid content, limiting the growth of some microbial species. Moreover, pottery jars are enclosed containers that only allow a trace amount of oxygen to pass through, while flavor compounds from wood barrels dissolve in wine, increasing the uncertainty of metabolic profile. For its relatively simple aging environment, huangjiu is suitable for machine learning modeling. In this study, we used huangjiu as an example of fermented alcoholic beverages to discuss the microbial influence on the metabolic profile. Through multiple targeted quantification methods combined with microbiome analysis, significant variables during the aging in pottery jars were identified. Machine learning models were constructed to illustrate the biomarkers that regulate the process of aging. Shapley additive explanations (SHAP) [27], a method that interprets the predictions of machine learning models through Shapley values [28], was applied to visualize the relationship between the microbiome and metabolome. The results showed that the aging year is the most powerful predictor for aging metabolic profiling and that the aging microbiome is associated with a variety of metabolites, indicating that the microbiome is an important regulatory factor for aging in pottery jars. Regulating microbial composition was a promising method to control the aging process of fermented alcoholic beverages.

## 2. Materials and Methods

### 2.1. Sample Collections

A total of 110 samples was obtained from the workshops of Zhejiang Guyuelongshan Shaoxing Wine Co., Ltd., (Shaoxing, China) for semi-dry huangjiu. The samples were collected from the bottom of the pottery jars. The supernatant was applied to measure physiochemical indexes and the metabolite profile of natural aging huangjiu, and the sediment (wine lees) was applied to measure the composition of the aging microbiome.

### 2.2. Measurement of Physiochemical Indexes

Five physiochemical indexes of aged huangjiu samples were measured, including total acid, amino nitrogen, reducing sugar, pH, and conductivity. Total acid and amino nitrogen were measured using acid-base titration. Reducing sugar was measured using the 3, 5-dinitrosalicylic acid (DNS) method. pH and conductivity were measured with a pH meter and conductivity meter, respectively.

### 2.3. Metabolite Profiling and Preprocessing

**Solid-phase microextraction–gas chromatographic–mass spectrometric (SPME-GC-MS).** Most volatile flavor compounds were quantified by SPME-GC-MS. An Agilent 8890 gas chromatographic with 7000D triple quadrupole GC/MS systems was applied. The supernatants of huangjiu samples (2 mL) were diluted with deionized water (4 mL). The mixture was then spiked with 2 g NaCl and internal standards (2-octanol, 100 mg/L, 10 μL) for further analysis. A VF-WAXms GC column (30 m × 0.25 mm inner diameter, 0.25 μm film thickness, Agilent Technologies, Inc., Santa Clara, CA, USA) was used to analyze the samples. Helium was used as the carrier gas at a rate of 1 mL/min. The initial oven temperature was 40 °C, held for 2 min, ramped to 150 °C at a rate of 4 °C/min, then ramped to 230 °C at a rate of 6 °C/min, and held for 5 min. The sample was desorbed for 5 min at 250 °C in splitless mode. The electron ionization mass spectra mode was used at 70 eV ionization energy. The temperature of the ion source was 230 °C. The temperature of quadrupole was 150 °C. The mass range was set from 35 to 350 amu in full scan mode.

**Liquid chromatography system (LCS).** Most amino acids were quantified by LC-MS. The supernatants of huangjiu samples (free from the protein) were diluted with deionized water and injected into an LCS (Agilent 1290 Infinity II LC system, Agilent Technologies, Inc., Waldbronn, Germany) through an AdvanceBio Amino Acid Analysis LC column (100 mm × 4.6 mm inner diameter, 2.7 μm film thickness, Agilent Technologies, Inc., Santa Clara, CA, USA). The column’s temperature was 45 °C. Mobile-phase liquid A was 10 mM Na_2_HPO_4_ and Na_2_B_4_O_7_ (pH = 8.20), and mobile-phase liquid B was mixture of methanol, acetonitrile, and deionized water (45:45:10 = v:v:v). Linear gradient elution (liquid A/%) was as follows: 0–0.35 min: 2%; 0.35–13.40 min: 2–57%; 13.40–13.50 min: 57–100%; 13.50–15.70 min: 100%; 15.70–15.80 min: 100–2%; and 15.80–18.00 min: 2%. The flow speed was 1.5 mL/min. The detector was diode array detector (DAD), the wavelength was 338 nm, and the bandwidth was 10 nm.

**Ion chromatography system (ICS).** Most organic acids were quantified by ICS. The supernatants of huangjiu samples (free from the protein) were diluted with deionized water and injected (20 μL) into an ICS (Dionex ICS-6000 Hybrid HPIC System, Thermo Fisher Scientific, Inc., Waltham, MA, USA) through the anion column (Dionex IonPacTM AS11-HC, 250 mm × 4 mm inner diameter, Thermo Fisher Scientific, Inc.) and the guard column (Dionex IonPacTM AS11-HC, 50 mm × 4 mm inner diameter, Thermo Fisher Scientific, Inc.). The column’s temperature was 30 °C. The leacheate was KOH solution generated by leacheate generator. The flow speed was 1.0 mL/min. Linear gradient elution (KOH concentration) was as follows: 0–16 min: 1.10 mM; 16–29 min: 1.10–16.50 mM; 29–35 min: 16.50–20.00 mM; 35–39 min: 20.00–35.00 mM; 39–41 min: 35.00 mM; 41–47 min: 35.00–50.00 mM; 47–47.1 min: 50.00–1.10 mM; and 47.1–59 min: 1.10 mM. The detector was electrochemical detector. The temperature of the detection cell was 35 °C.

**ICS.** Most monosaccharides were also quantified by ICS. The supernatants of huangjiu samples (free from the protein) were diluted with deionized water and injected (20μL) into an ICS (Dionex ICS-6000 Hybrid HPIC System, Thermo Fisher Scientific, Inc.) through the ion column (CarboPac PA20, 150 mm × 3 mm inner diameter, Thermo Fisher Scientific, Inc.) and the guard column (CarboPac PA20, 50 mm × 3 mm inner diameter, Thermo Fisher Scientific, Inc.). The column’s temperature was 30 °C. Mobile-phase liquid A was deionized water, and mobile-phase liquid B was 250 mM NaOH. The flow speed was 0.50 mL/min. Linear gradient elution (liquid A/%) was as follows: 0–20 min: 99%; 20–20.1 min: 99–20%; 20.1–25 min: 20%; 25–25.1 min: 20–99%; and 25.1–30 min: 1%. The detector was sulfur chemiluminescence detector.

**Data preprocessing of metabolomics data.** Metabolites with less than 50 measurements were removed from the datasets. We regressed metabolite levels with sample storage time and imputed missing values as the minimum value per metabolite. We performed z-score standardization over log (base 10)-transformed level of metabolites for microbiome modeling.

### 2.4. Sequencing and Microbiome Preprocessing

Total DNA was extracted from rice wine lees using cetyltrimethylammonium bromide (CTAB) method. The 16S rRNA gene was amplified with universal primers 27F_NGS (5′-AGRGTTTGATYNTGGCTCAG-3′) and 1492R_NGS (5′-TASGGHTACCTTGTTASGACTT-3′) with the barcode, and the purified amplicon pool PCR products were applied to construct the SMRT Bell sequencing libraries. Sequencing was performed using the PacBio Sequel II instrument.

All clean CCS sequences were clustered into amplicon sequence variants (ASVs) and assigned to genus and species levels. Alpha diversity evaluations of Chao1, Shannon, and Simpson diversity indices were calculated to understand the microbial richness and diversity.

Species or genera with more than 0.1% average relative abundance and more than 10 measurements were kept for further analysis. Bacterial relative abundance data were transformed with a centered log-ratio transformation (zero scores were replaced with 10^−10^) for machine learning modeling.

### 2.5. Predictive Models of Aging Metabolites

To compare the performance of linear models with tree-based models in processing natural-aging huangjiu data, explained variance and coefficient of determination of every single metabolite obtained from the predicted level of linear model and tree-based model were applied to evaluate. Lasso regression (scikit-learn 1.0.2) was applied as the linear model. LightGBM (lightgbm 3.2.1), XGBoost (xgboost 1.5.0), and random forest were applied as the tree-based models. All of the models were performed in five-fold cross-validation, where in each fold, we ran a hyperparameter search consisting of 10 iterations. Each model was run 10 times to avoid overfitting.

Feature groups for further modeling included physiochemical indexes (total acid, amino nitrogen, reducing sugar, pH, conductivity, and aging year) and microbial data (diversity indexes and relative abundance). Feature groups with different compositions were applied for modeling. The names of feature groups and the composition of feature groups were as follows: (i) full model (FM): all of the feature groups; (ii) physiochemical indexes (PI): all of the physiochemical indexes; (iii) aging-related factors 1 (AFS): aging year and microbial data annotated into species; (iv) aging microbiome 1 (AMS): microbial data annotated into species; (v) aging-related factors 2 (AFG): aging year and microbial data annotated into genera; and (vi) aging microbiome 2 (AMG): microbial data annotated into genera.

We then implemented random forest models in scikit-learn package to evaluate the performance of different feature groups in predicting metabolite levels. In order to estimate the explained variance of metabolite groups, we used each feature group as input and constructed fivefold cross-validation models. We calculated 95% confidence intervals and *p*-values of the predicted results through 100 bootstrapping iterations. In each bootstrap iteration, we randomly performed five-fold cross-validation, and in each iteration, we randomly replaced a group of datasets from the training set to have the same size as the current training set. We then used this dataset to train our model and evaluate model performance in the remaining fold. Finally, we calculated the coefficient of determination between measured metabolite levels and cross-validated predicted levels obtained from bootstrapping iterations.

A set of fixed parameters were applied in the machine learning pipeline: n_estimators: 100; criterion: ’Gini’; max_depth: 5; min_samples_leaf = 5; min_samples_split: 2; and max_features:10.

### 2.6. Feature Attribution Analysis

To explain the output of the machine learning model and further reveal the relationship between aging-related factors and metabolites, we applied SHAP, a visualization tool that can explain the output of machine learning models. Shapley-value-based analysis has been shown to be useful in gut microbiome data in recent years.

In every five-fold cross-validation, module TreeExplainer (v 0.39.0) [29] was applied to calculate individual SHAP values. As standardizations were applied in metabolite groups and microbiome data, SHAP values would be comparable in models of different metabolites.

Mean absolute SHAP values were calculated to reflect the average influence of every feature on metabolite levels. Then, mean absolute SHAP values multiplied by the sign of Spearman correlation between features and metabolite levels were taken as directional mean absolute SHAP values. Positive values lead to higher metabolite levels, while negative values lead to lower metabolite levels on average.

### 2.7. Bioinformatics and Statistical Analysis

The original data obtained from sequencing was further processed with QIIME 2 (2021.2) [30]. The sequence set was denoised and clustered into ASVs using DADA2. The single representative sequence was aligned with the Silva database. The Chao1 richness index, Shannon diversity index, and Simpson diversity index were calculated using QIIME 2 (2021.2) [30].

For all statistical analysis and predictive models, we used Python 3.7.13 with the following packages: pandas 1.3.5, numpy 1.21.5, scikit-learn 1.0.2 [31], lightgbm 3.2.1 [32], xgboost 1.5.0, and shap 0.39.0 [27].

## 3. Results

### 3.1. An Aging Huangjiu Microbiome-Metabolomic Dataset Collection

We collected 110 huangjiu samples stored in pottery jars with aging years of 3, 8, 10, 20, and 25 to investigate the factors related to aging and quality of huangjiu. We measured the physiochemical indexes (Figure 1) and used HPLC-MS, IC, and SPME-GC-MS to profile metabolite composition. Over 100 metabolite levels were measured using our methods, covering a wide range of common substances including organic acids, monosaccharides, amino acids, alcohols, esters, and aldehydes (Appendix A). We quantified 74 metabolites and included these metabolites for further analysis after quality control. Total acid is an important index to evaluate the quality of aging huangjiu. High acidity leads to spoilage and even undrinkable quality. Most physiochemical indexes and metabolite levels showed significant correlations (Spearman *p* < 0.05) with total acid, which means changes in total acid were likely linked to the quality of huangjiu.

We also collected wine lees for PacBio full-length high-throughput sequencing. According to the database alignment results, aging microbiome from wine lees belonged to 16 bacterial phyla. *Firmicutes*, *Proteobacteria*, *Bacteroidetes*, and *Actinobacteria* were the main populations detected in aging huangjiu samples. At the genus level, 282 bacterial genera were involved in all the samples (Figure 2). Based on the average relative abundance detected in all samples, *Pseudomonas* was the most abundant bacterial genera, accounting for 59.16% on average. Furthermore, *Ralstonia*, *Methyloversatilis*, and *Cupriavidus* were also abundant in aging huangjiu samples, which were discussed less in huangjiu production. LAB such as *Peribacillus* (4.22%), *Lacticaseibacillus* (1.64%) and *Lactobacillus* (0.98%) were also detected in the partial samples. The base wine would be sterilized before being sealed into pottery jars to remove the majority of microorganisms accumulated during the fermentation process, which resulted in the low abundance of LAB.

Ranking huangjiu samples according to total acid, the relative abundance of LAB was on the rise, and *Pseudomonas* was on the decrease, with total acid increasing (Figure 2a). Aging year is another key factor that influenced microbial composition. Though it was the main producer of organic acids, LAB did not exist in some huangjiu samples with high acidity. Some microbial genera could disappear in the microbial succession of long-term storage. We then ranked huangjiu samples according to aging year (Figure 2b). In 3-year-aged huangjiu, the microbial genera were relatively abundant compared to high-aging-year samples. Notably, *Methyloversatilis* showed a significant positive relationship with aging year. As for alpha-diversity indices, Chao1, Shannon, and Simpson indexes all showed a weak correlation with total acid and aging year (|Spearman *r*| > 0.1). Thus, total acid and aging year were chosen as the primary indexes for further analysis.

### 3.2. Construction of Aging Huangjiu Microbiome-Metabolome Models

We constructed tree-based models and linear models to predict metabolite levels in aged huangjiu. As linear methods such as canonical correlation analysis (CCA) and redundancy analysis (RDA) were usually used in the analysis of fermented microbiomes, we used a Lasso regression model to compare the accuracy and efficiency of linear models with tree-based models. As a result, LightGBM, XGBoost, and random forest all outperformed the linear model. Random forest predicted most metabolites (29 ± 1 in 10 runs), while Lasso only predicted 19 ± 1 metabolites (Figure 3a). The explained variances of Lasso regression were also significantly lower than those of tree-based models (Figure 3b). As the sample size in our study is 110, some of the machine learning models were more suitable for a larger dataset. Considering differences and the median in coefficient of determination, random forest was selected for further modeling to avoid overfitting (Figure 3c–f).

In total, 64 of the 74 metabolites were significantly explained by PI, which means these physiochemical indexes well reflected the quality of aged huangjiu. Notably, 31 metabolites were significantly explained by the aging microbiome. Our models explained over 10% explained variance of 53 metabolites (median 20.7%, maximum 79.1%, Figure 3c), and more than 50% explained variance of 15 metabolites.

We next checked the prediction performance in different types of metabolites (Figure 4d). FM predicted well for most organic acids, monosaccharides, esters, and aldehydes. As for aging-related factors, the most well-predicted metabolites in AFS could be predicted by PI. None of the models performed well in amino acids. Organic acids, alcohols, esters, and aldehydes were all important flavor compounds of huangjiu and underwent complex transformations during aging. Understanding the rules of metabolite change helps to regulate and control the quality of aging huangjiu.

### 3.3. Index–Metabolome Association of Aging Huangjiu

To further reveal the relationship between physiochemical indexes and metabolite levels, a cluster map was created with 31 well-predicted metabolites (explained variance > 0.3) to describe the variations (Figure 5). As illustrated in the cluster map, well-predicted metabolites were divided into three clusters. MC-1 included most organic acids and some esters and aldehydes; MC-2 included most of the flavor substances; and MC-3 included amino acids and aldehydes. Many known aging biomarkers of huangjiu were involved in MC-2, such as vanillin, acetophenone, and benzyl alcohol. The differences in metabolite levels led to three clusters of huangjiu samples. SC-1 was the highest-quality huangjiu cluster, with more flavor compounds and moderate organic acids.

Different clusters of huangjiu samples showed a significant variation in total acid (one-way ANOVA *p* < 0.01). Flavor substances were low in huangjiu of low acidity (SC-3) and high in huangjiu of medium acidity (SC-1), while a drop in these metabolites was observed when acidity continued increasing (SC-2). A medium-acid environment was likely to promote reactions leading to flavor substance formation, such as the esterification reaction. Twenty-eight of the thirty-one well-predicted metabolites also showed a significant correlation with total acid (|Spearman *r*| > 0.3, Spearman *p* < 0.01), indicating that total acid is an important factor to regulate and control the aging process.

### 3.4. Influence of Aging-Related Factors on Metabolites

Though the effect of Lactobacillus on lactic acid was illustrated by previous studies, the metabolic properties of other aging bacteria were not investigated yet. We compared the explained variance of metabolites obtained by models of aging-related factors to reveal the relationship between bacteria and metabolites. Species-level and genus-level annotation had similar explained variance. When aging year, a known aging-related factor, was added to the model, model performance improved for 86% of the metabolites (Figure 6). These metabolites included several previously known aging markers. Some of the metabolites did not predict well in AMS but predicted well in AFS (hexanal, acetoin, 5-methylthiophene-2-carboxaldehyde, etc.), indicating that time-related physiochemical reactions were the main factors in these cases. For well-predicted metabolites that improved prediction performance in AFS (4-ethylphenol, 1-octen-3-ol, etc.), time-related physiochemical reactions and microbial activities together drove these formation reactions in long-term aging. Some of the metabolites did not predict better when aging year was added to the model, such as lactic acid and ethyl 2-hydroxy-4-methylvalerate. Microbial activities could be the driving forces behind formation of these metabolites. The succession of related microbial communities reached a stable status in relatively short time, so microbial compositions were able to predict these metabolites.

### 3.5. Contribution of Aging Microbiome

To interpret AFS models and infer the microbial driving forces of different metabolites, we used feature attribution analysis (SHAP) to visualize the prediction result (Figure 7g). Though different indexes could not be compared together, aging year was still the most powerful contributor to aging metabolites. Various aging biomarkers that were verified by previous studies also show correlations in our result (Figure 7a–c) [4,15]. Furthermore, alpha-diversity indices were not strongly predictive of aging metabolites.

Apart from aging year, most of the metabolites still had at least one strong predictive factor of species (mean absolute SHAP >0.1). Lactic acid was proven to be the main cause of spoilage in huangjiu, and *Lactobacillus* was the main generator of lactic acid during huangjiu aging. *Lactobacillus acetotolerans* was a strongly predictive factor for lactic acid in our study, as previously reported (Figure 7d) [10,33]. Apart from lactic acid, *Lactobacillus acetotolerans* was also a positive predictive factor for many esters and organic acids. While some of these metabolites could be produced by *Lactobacillus acetotolerans*, the formation of other metabolites was possibly driven by a medium-acid environment. Some relationships were not investigated in the previous study, such as those between *Peribacillus frigoritolerans* and ethyl 2-hydroxy-4-methylvalerate (Figure 7f). Significantly, *methyloversatollis discipulorum* showed a similar mean absolute SHAP value with aging year (Spearman *r* = 0.834, Spearman *p* < 0.001, Figure 6e). However, *Methyloversatollis discipulorum* was usually discovered in the environment, and there was no evidence show it had any effect on aged huangjiu. Smalley et al. [34] isolated *Methyloversatollis discipulorum* from Lake Washington and proved its metabolic capabilities of utilizing organic acids, alcohols, and aromatic compounds. Moreover, despite being on rise with aging, *Methyloversatollis discipulorum* did not exist in several aged huangjiu samples. Thus, the metabolic properties of *Methyloversatollis discipulorum* during aging needed more experimentation to be verified.

Inferring microbial correlations is crucial to understanding the mechanisms of microbial communities. However, these correlations are dynamic and depend on environmental conditions. The initial properties of huangjiu and the aging environment are identified as key factors in the establishment of a microbial community. In this study, the aging environment (alcohol and oxygen content) of huangjiu restricted microbial species. The metabolic properties during aging may differ from those in the fermentation process. The introduction of fermentation microorganisms during aging required metabolism tests to further analyze the interactions. Furthermore, whether some of the environmental microorganisms are identified as functional microorganisms and can be applied in huangjiu should be considered cautiously. A comprehensive analysis of the aging microbiome helps to prevent spoilage during aging and promote the aging process to some extent.

## 4. Discussion

**The effectiveness of machine learning and explainable artificial intelligence methods in the aging models**. In our study, machine learning methods were shown to be an effective data mining method for multi-omics data on fermented foods. Although the dataset was not large enough to fully reveal the microbial driving forces for aging, our study was the first to combine the aging microbiome with targeted metabolomics by using machine learning methods in the analysis of aged huangjiu. Several important flavor compounds were significantly predicted by our models, verifying the effectiveness of machine learning and explainable artificial intelligence methods in aging models. However, there were several limits to our methods. First, because of the limitations of sample amounts and detection methods, many factors were ignored in our studies, which may lead to deviations in the prediction of some metabolites. Second, though previous studies could support some of the results, most correlations in our study should not be considered causal and need more experiments to verify. Third, on account of the lack of reliable annotations, we did not associate metabolites with specific enzymes. Subsequent studies could focus on enzymes to regulate the aging process. Finally, because our models were not validated with external datasets, some results could differ when the environment changed.

The effectiveness of machine learning in aging models indicates that the methods can be extended to other production processes for fermented alcoholic beverages. Microbial community succession occurs with the dynamic change of physiochemical indexes in the fermentation process after the introduction of the wine starter. Experienced brewers regulate the fermentation process by controlling environmental factors, which leads to quality differences in different branches. More data are available for building a fermentation model with the help of Internet of Things monitors. In the big data era, machine learning methods tend to play an important role in regulating the production of fermented alcoholic beverages, including the fermentation process and the aging process.

**Positive microbial influence on the aging microbial profile of fermented alcoholic beverages**. As mentioned above, plenty of studies have been conducted on the metabolic profile and physiochemical reactions of natural aging huangjiu, so the relationship obtained by this study between physiochemical indexes is easy to verify. However, with regard to the aging microbiome, previous research has focused more on microorganisms in huangjiu. In our study, we collected the sediment (wine lees) of centrifugated huangjiu for microbiome analysis, resulting in different microbial compositions. More environmental microorganisms were observed in wine lees (*Pseudomonas*, *Ralstonia*, and *Methyloversatilis*), while fermentation-related microorganisms predominated in huangjiu (mainly LAB) [10,33]. Many environmental microorganisms influence the quality of huangjiu because the production process is open fermentation and allows the fermentation broth to come into direct contact with the fermentation environment [35,36]. These environmental microorganisms do not exist in stainless-steel vessels. The relationship between metabolic profile and environmental microorganisms found in our study could be one of the key factors that causes the differences in different containers.

Though the model in our study is relatively simple, as the microbial succession occurs smoothly during aging, the observed and predicted results were quite credible. Moreover, the results could be generalized to other fermented alcoholic beverages by adding new predictive factors. Aging microorganisms were commonly considered to be harmful in fermented alcoholic beverages. LAB and AAB that are derived from fermentation are commonly associated with spoilage. In our study, moderate abundance of some LABs improved the composition of flavor compounds. Additionally, some microorganisms that came from aging containers and environment strongly correlated with metabolic profiles. Many of these correlations showed that aging microbiomes promote the aging process of fermented alcoholic beverages. Though many of the flavor compounds of aging wine and beer come from wood barrels, the microbial influence cannot be easily ignored. Fermented alcoholic beverages such as beer and wine are better habitats for microorganisms than huangjiu. Thus, though the aging time is short, the aging process will have a more dramatic microbial influence on the metabolic profile. Understanding microbial succession and function helps to regulate the aging process of fermented alcoholic beverages.

**Potential biological methods to accelerate aging.** In the traditional production process of fermented alcoholic beverages, certain wine containers are used for aging. For example, wine and beer are stored in different types of wood barrels, and huangjiu are sealed into pottery wine jars (about 20 L) with mud for aging. Both wood barrels and pottery jars are permeable to air, providing tiny amounts of oxygen for transformation during aging. The difference is that pottery jars are rich in metal ions, which accelerate aging-related reactions including esterification, redox, and condensation [37,38,39], while flavor compounds from different wood barrels change the composition of metabolites in beer and wine. Stainless-steel vessels have the advantages of corrosion resistance, large capacity, and low loss, making them suitable for large-scale storage of fermented alcoholic beverages in industrial production. However, its structural difference restricts tiny oxygen flows and affects the aging process. In addition, stainless-steel vessels release heat slowly due to their large capacity, which may lead to deterioration and taste abnormalities. Out of these shortcomings, the mechanisms of compound transformation in traditional aging containers were investigated to cover the differences, and various artificial aging technologies were developed, such as microwave, electric field, high pressure, and ultrasonic [40,41,42]. However, there are problems with these commonly used physical and chemical methods, limiting their use in practical production.

Biological methods tend to be ideal methods for aging of fermented alcoholic beverages. Iosip et al. [43] used yeast derivatives for aging. Because microorganisms are usually undesirable in the aging process, the enzymes that are generated by aging microorganisms are more suitable for adding to fermented alcoholic beverages and easier to control during the aging process. Understanding the metabolic pathway and screening for suitable enzymes in aging microbiomes are necessary for further research. Immobilized enzyme reactors represent the potential application of our results in the practical production process of fermented alcoholic beverages.

## 5. Conclusions

In summary, our results illustrate the relationship between physiochemical indexes and huangjiu quality. Total acid was the most important index that must be controlled. In addition, analyses of intercorrelations showed a large number of significant intercorrelations between individual indexes and metabolites. Potential microbial determinants for metabolites of natural aging huangjiu were also revealed through SHAP values. Many of the associations found in this study replicate previously reported findings, demonstrating the efficacy of machine learning methods in multi-omics analyses of fermentation-related microbiomes and metabolomes. The majority of them are new, making them a useful resource for further controlling and regulating the quality of huangjiu and other fermented alcoholic beverages during aging. It needs to be emphasized that all these results cannot be simply deduced from the direct changes in metabolites and microbial composition. The complexity of the interactions always has to be taken into consideration in further validation experiments.

## Figures and Tables

**Figure 1 foods-12-00906-f001:**
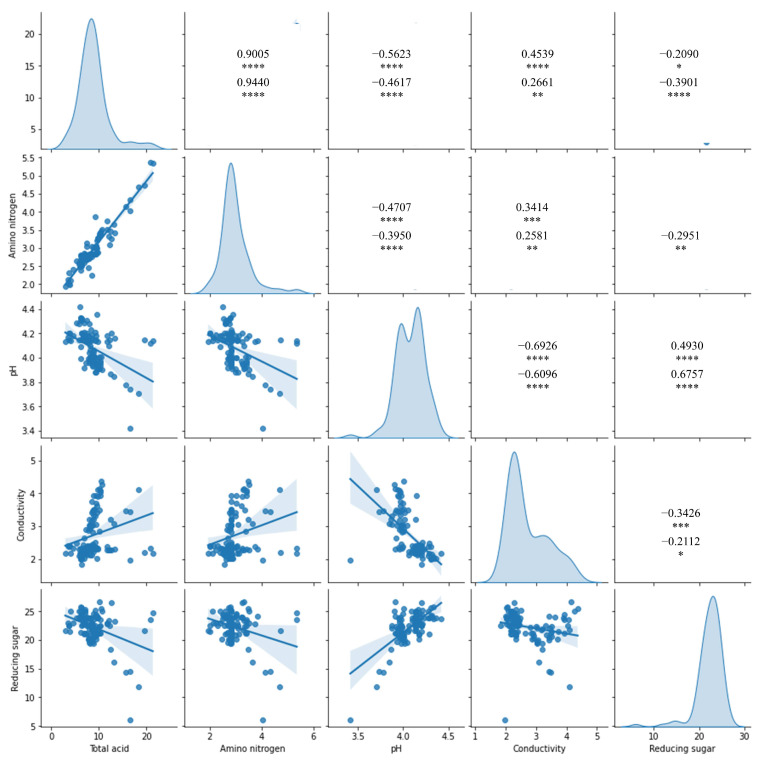
Dot plots of the physiochemical indexes of age huangjiu samples. Panels on the diagonal show the distribution of physiochemical indexes. Figures in the top right are Spearman ρ, Spearman *r*, Pearson *p*, and Pearson *r* from top to bottom, respectively. *: *r* < 0.1; **: *r* < 0.05; ***: *r* < 0.01; ****: *r* < 0.001.

**Figure 2 foods-12-00906-f002:**
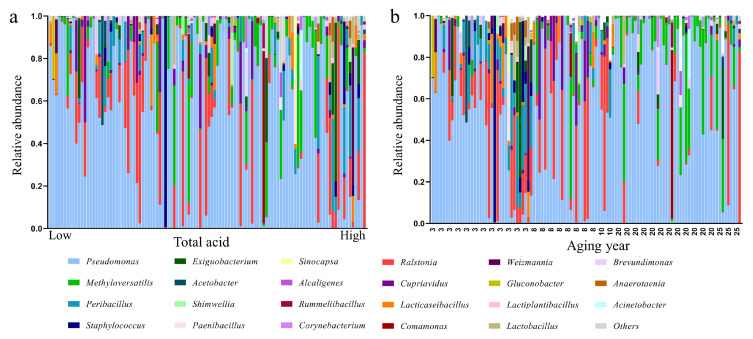
Distribution of genus of aged huangjiu samples. (**a**) Rank according to total acid. (**b**) Rank according to aging year.

**Figure 3 foods-12-00906-f003:**
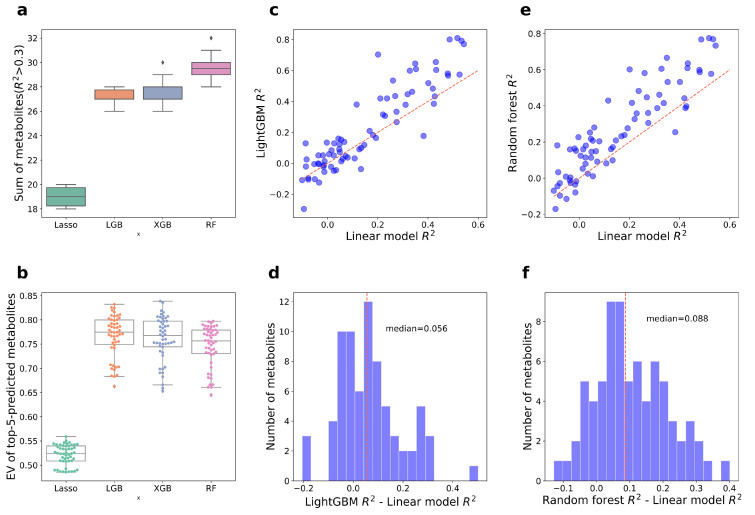
Model selection. (**a**) Sum of metabolites (*R*^2^ > 0.3) for different models. (**b**) Box and swarm plot of explained variance of top five predicted metabolites in 10 runs. (**c**,**e**) Dot plots of the explained variance of the metabolite groups from different machine learning models. (**d**,**f**) Distribution of difference value between different models.

**Figure 4 foods-12-00906-f004:**
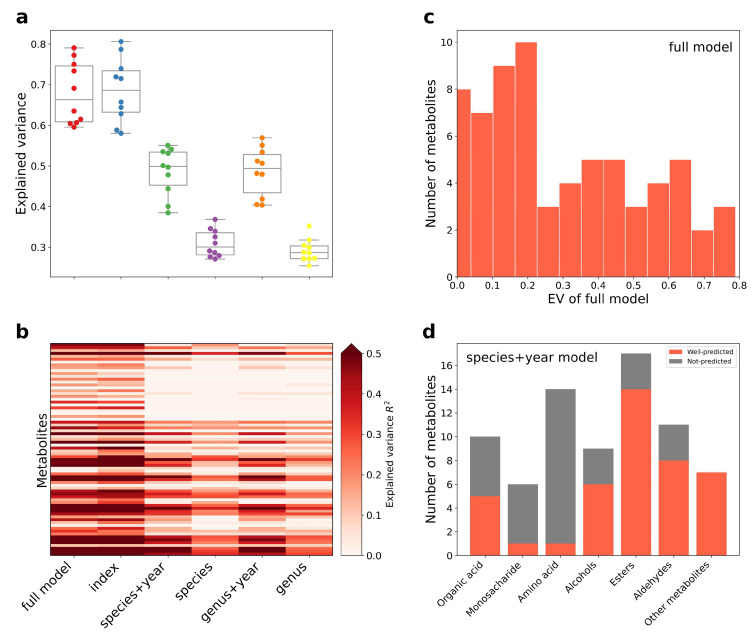
Model performance in feature groups. (**a**) Box and swarm plot of explained variance of top 10 well-predicted metabolites of each model. (**b**) Heatmap of explained variance for each metabolites of different models. (**c**) Distribution of explained variance of metabolites for FM model. (**d**) Prediction performance in different metabolite group.

**Figure 5 foods-12-00906-f005:**
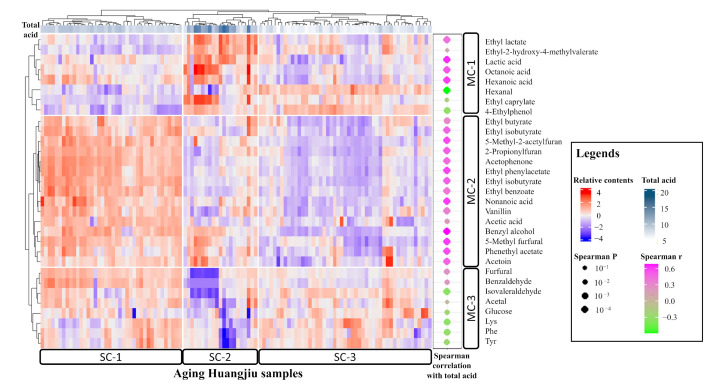
Cluster map of relative contents of metabolites that were well predicted by PI model (explained variance > 0.3). MC: Metabolite Cluster; SC: Sample Cluster.

**Figure 6 foods-12-00906-f006:**
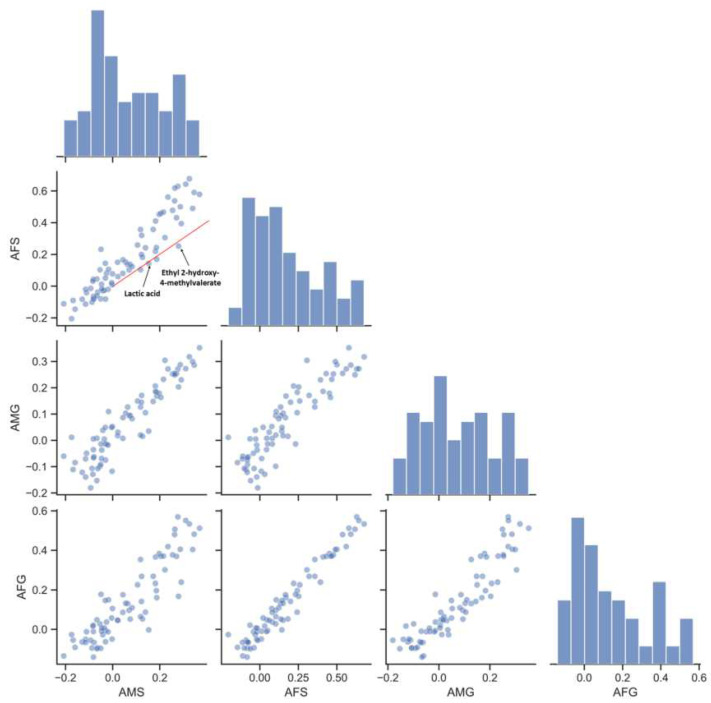
Comparison of explained variance of metabolites for aging-related models. Dot plots of the explained variance of the metabolite groups from models based on every pair of age-related models. Panels on the diagonal showed the marginal distribution of explained variance of metabolite groups for a certain age-related group. Aging-related factors 1 (AFS): aging year and microbial data annotated into species; aging microbiome 1 (AMS): microbial data annotated into species; aging-related factors 2 (AFG): aging year and microbial data annotated into genera; and aging microbiome 2 (AMG): microbial data annotated into genera.

**Figure 7 foods-12-00906-f007:**
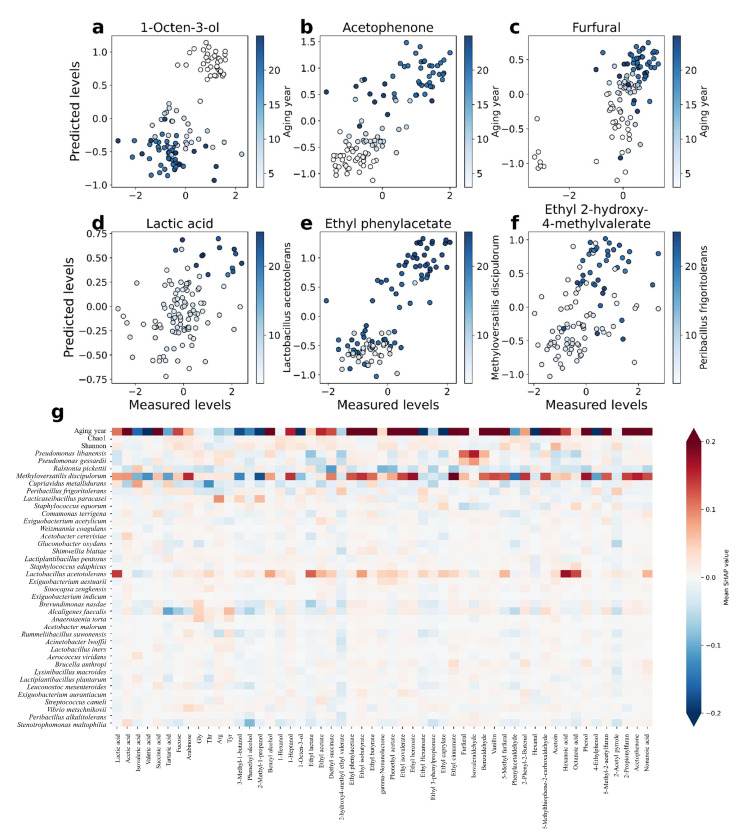
Specific bacterial taxa underlie the accurate prediction of aging metabolites. (**a**–**f**), predicted (y-axis) versus measured level of 1-octen-3-ol (**a**), acetophenone (**b**), furfural (**c**), lactic acid (**d**), ethyl phenylacetate (**e**), and ethyl 2-hydroxy-4-methylvalerate. Predictions of a–c colored by the aging year. Predictions of d–f colored by the standardized relative abundance of bacterial taxa. (**g**) Heat map shows the directional mean SHAP values of aging-related factors.

## Data Availability

The data are available from the corresponding author.

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
