# Peer review of "Impact of Aging Microbiome on Metabolic Profile of Natural Aging Huangjiu through Machine Learning"

_foods, 2023, doi:10.3390/foods12040906_

Round 1
Reviewer 1 Report
Authors should check the format of the references
Reviewer 2 Report
The research article “Impact of aging microbiome on metabolic profile of natural aging huangjiu through machine learning” represent an interesting work.
Please consider the following recommendations for the introduction section: Precise, Comprehensive, Need, why, Background, and History.
The introduction section misses the most recent literature review. Please consider adding the most recent studies related to the given problem. Further, irrelevant information must not be added in this section.
Please consider the following recommendations in the result and discussion sections: how these results could impact? how these results could be increased? and what future direction could be taken to enhance the methodology? i.e., add recommendations, future aspects, etc. sections.
The English language in the manuscript is not up to mark. Please consider taking professional editing services to remove grammatical errors and spelling mistakes.
Few comments are:
Abstract part needs to be improved
Line 37: Remaining instead of remining
Line 85: Chen et al[28] >>>>>>> it should be Chen et al. [28]
Line 88: Bar et al[29] >>>>>>> it should be Bar et al. [29]
Line 135-139: 230 °C >>>>>>>>>>>> it should be 230°C. (no space between numerical value and degree centigrade symbol)
Line 164: quantified by ICS. he supernatants????????????
Line 421: verifying the the effectiveness of machine>>>>>>> why repeated words
Conclusion part needs to be improved
Too many mistakes in the reference section, please check the guidelines of the journal regarding references and stick to it.
· Somewhere you have used full name of journal and somewhere you have used abbreviated form. It should be same.
· If you are writing years in bold, it should be the same.
· Why are you using et al in the references? Write all names of authors
Reviewer 3 Report
The topic is quite original and adds more to machine learning and correlates with fermentative microorganisms.
The manuscript is scientifically sound and the experimental design is appropriate to test the hypothesis ( a large no. of samples along with metabolite profiling techniques aqnd modeling of metabolomics)
The results reproducible based on the given methods.
All data presented (figures, tables) is interpretable and understandable.
Conclusions are consistent with findings.
Cited references are appropariate and recent.
Very good research paper however English editing is required at cases.
e.g. To explore the factors that related to aging and quality of huangjiu, we collected 110 251 huangjiu samples stored in pottery jars.
We trained tree-based models (constructed)
we taken 296 Lasso regression model (used)
line 37 remining 5% (remanining)
Round 2
Reviewer 2 Report
Manuscript has been revised